# OpenReview forum: "CodeBiasBench: Benchmarking social fairness of large language model generated code"
_ICLR.cc/2026/Conference — Submitted to ICLR 2026_

### Official Review · Reviewer_Pqhm · 2025-10-20

**Soundness:** 1
**Presentation:** 2
**Contribution:** 1
**Rating:** 2
**Confidence:** 4

**Summary:**

The paper introduces CodeBiasBench, a benchmark for evaluating social fairness in code completion using designed templates with sensitive parameters (e.g., age, gender). It further proposes a Contrastive Chain-of-Thought detection procedure that compares completed functions under different sensitive-attribute settings by creating assertions. The experimental results reveal that many models exhibit fairness disparities, especially the bias scores on the Neutralized (template) subset exceed those on the Sensitive subset, which the authors interpret as implicit bias.

**Strengths:**

1. This paper provides an unexplored way to evaluate social fairness in code completion compared to text generation fairness.
2. They evaluated multiple LLMs, showing an effort to benchmark fairness.

**Weaknesses:**

1. The validity of the proposed completion tasks is questionable.
This paper does not verify whether the designed code completion tasks are valid for evaluating intrinsic model bias. In the benchmark template in Figure 3, it explicitly embeds sensitive parameters (e.g., age, gender) and tasks the model to just fill in the missing conditions without any specific requirements, defined target, or ground truth. I have a concern that this setup solely forces models to manipulate these variables regardless of functional relevance, letting the task become a form of syntactic pattern completion rather than bias detection.  It is unclear whether the observed disparities or completed code reflect social bias or merely artifacts of the template construction. Although the authors designed the Neutralized Subset, the template still has these concerns (see Weakness 2). The lack of ablation analysis and additional experiment verification leads this study to overclaim bias violations observed in their experiments.

2. Invalid control-group design.
The core experimental comparison between the Sensitive and Neutralized subsets is not sound. The Neutralized subset is meant to remove all sensitive cues so that any differences in completions could be attributed to implicit bias. However, your Neutralized templates still retain the sensitive parameters in the function parameters, intentionally miss the conditions, and let LLMs "complete" (providing five parameters with five conditions, then asking models to complete), encouraging models to use that "sensitive information" in the completion context and contaminating the control condition. Two subsets provide similar semantic cues about the sensitive parameters, making any observed difference statistically meaningless. In Section 4.2.1, the models show higher bias scores on the Neutralized subset, which indicates that the evaluation is invalid. However, the authors interpret it as a stronger "implicit bias", which is logically inconsistent. I think this benchmark cannot reliably measure what it claims to evaluate.

3. Lack of fairness specification or ground-truth definition.
As mentioned in Weakness 1, this paper does not define what constitutes a "fair" or "unfair" completion. The authors conflate random variation or stylistic diversity with true discrimination, because any difference between completions is arbitrarily labeled as "bias". This omission makes the CBS scores uninterpretable and not reproducible.

4. Circular evaluation using LLM-generation and LLM-based detection.
Both the completion (ask LLMs to complete the benchmark templates) and detection (use CCoT to judge whether the function is biased) are dependent on LLMs. This LLM-to-LLM feedback loop introduces self-confirmation bias, with no independent ground for validation.

**Questions:**

1. Could you clarify how you distinguish between an "arbitrary completion" and a "genuinely biased completion"? In other words, how do you define a "fair" and "unfair" completion?
2. The Neutralized subset still contains those sensitive parameters. How can you ensure that the different results from the two subsets are not template artifacts?

Minor suggestions:
1. The \vspace usage is too obvious in page 7 and 9, which violates ICLR style requirements. I will not suggest a desk rejection but will urge you to modify accordingly.

---

### Official Review · Reviewer_kUAR · 2025-10-26

**Soundness:** 2
**Presentation:** 2
**Contribution:** 2
**Rating:** 2
**Confidence:** 4

**Summary:**

This paper introduces CodeBiasBench, a benchmark intended to evaluate social bias in LLM code completion. The dataset contains a Sensitive subset (contains conditions related to sensitive attributes) and a Neutralized subset (removes all conditions associated with sensitive attributes). The authors define a metric, Code Bias Score (CBS)—the proportion of generated code functions deemed biased—and propose a detection method called Contrastive Chain-of-Thought (CCoT) that employs contrastive reasoning to analyze the generated code and identify potential bias. Experiments measure CBS for several models (GPT-3.5-Turbo, GPT-4, DeepSeek-Coder variants, Qwen2.5, CodeLlama), with GPT-4o used as the detector in zero-shot CCoT. The paper claims that LLMs show substantial bias across age, region, gender, and race, and argues that Sensitive/Neutralized formats better reflect 'realistic' code completion while avoiding ethical safeguard triggers.

**Strengths:**

1. Meaningful topic with practical value (i.e., bias in code generation).
2. Evaluating both explicit and implicit biases.

**Weaknesses:**

1. Inappropriate wording and confusing paper organization.
2. Missing necessary details in dataset construction and prompt design.
3. Outdated models and missing recent literature.
4. Typos and presentation issues.

**Questions:**

1. **Inappropriate wording**: In line 90, the authors claim that `our focus is on more practical code completion scenarios`.  How did they collect and design `more practical scenarios`? What are the sources of scenarios like job hunting? I understand that the authors want to express that this paper focuses on coding tasks in practical scenarios, but could they provide any evidence to prove that the scenarios in this work are **more** practical (than the prior work)?
2. **Missing necessary details**: (1) In Line 201, the authors mention that details of original code construction are in Appendix A, but Appendix A provides only a cursory description and lacks details. There are many crucial details about the validity and rationality of the dataset to be supplemented: How did the authors design the code template? Did the design refer to existing work or technical reports? How did they effectively select from the hundreds of candidate codes? What are the selection criteria? The paper lacks the necessary details about the dataset construction and also does not provide any explanation in the supplement materials. (2) One key motivation for this paper is that existing research is prone to triggering the LLM safeguard mechanism. However, the authors do not provide the reference or observations (with concrete numbers) to support or demonstrate the severity of this issue, which reduces the authenticity and urgency of this motivation.
3. **Confusing paper organization**. Section 3 introduces the main methods of this paper, while Section 3.2 discusses several bias detection methods (including CCoT), and Section 3.3 separately introduces the CCoT method. So are other bias detection methods part of the detection method of this paper? If not, Section 3.2 should be moved to the background or related work which talks about the existing methods. In this case, CCoT should be introduced directly after Section 3.1. If zero-shot and other detection methods are part of the methods of this paper, then Section 3.2 and Section 3.3 should be merged, and CCoT should be part of `Bias Detection`, rather than a parallel section.
4. **Missing recent references**. LLM code generation is a rapidly evolving field, but there is no reference from 2025 and only 4 papers from 2024 in this paper. Most of the references have been published for two or even three years. Authors should include and discuss more recent papers about the social bias in LLM code generation, e.g., [1][2][3].
5. **Typos and presentation problems**: For example, (1) 'Different LLMS were detected' in Figure 1, should be 'LLMs'; (2) 'Appendix Band C' in Section 3.3, should be 'Appendix B and C'. These typos reduce clarity and credibility.


[1] Faircoder: Evaluating social bias of llms in code generation, 2025.

[2] Software Development Life Cycle Perspective: A Survey of Benchmarks for Code Large Language Models and Agents, 2025.

[3] Bias unveiled: Investigating social bias in LLM-Generated Code, AAAI 2025.

---

### Official Review · Reviewer_F5Be · 2025-11-03

**Soundness:** 2
**Presentation:** 3
**Contribution:** 2
**Rating:** 4
**Confidence:** 4

**Summary:**

This paper proposes CodeBiasBench, a new benchmark to evaluate social fairness in code generated by large language models (LLMs), focusing on code completion tasks. Unlike prior work that mainly uses natural language or synthetic snippets, CodeBiasBench constructs over 5,000 template-based tasks that simulate realistic code completion conditions. The authors also introduce Contrastive Chain of Thought (CCoT), a bias detection method that uses contrastive reasoning to compare model outputs under different sensitive attributes (e.g., age, gender, race). Experiments are conducted on several open- and closed-source models (e.g., GPT-4, DeepSeek-Coder, Qwen2.5, CodeLLaMA), with the proposed Code Bias Score (CBS) quantifying the extent of bias. The authors find that all tested models show measurable bias and that CCoT performs better than zero-shot and few-shot baselines in detecting bias.

**Strengths:**

+ Addresses an underexplored area: The topic of fairness in code generation is still quite new, so having a dedicated benchmark is a meaningful step forward.

+ Methodological novelty: The proposed CCoT approach adds an interesting contrastive reasoning component that seems to improve detection consistency and efficiency.

+ Comprehensive experiments: The authors evaluate a variety of model families and versions, giving the results more coverage.

**Weaknesses:**

- Questionable realism of the benchmark
- Reliance on LLM self-assessment
- Lack of clear implications

While fairness is an important general topic, the paper doesn’t convincingly argue why fairness in code generation is an urgent or impactful problem. It’s not fully explained how bias in code completion actually causes harm or what kind of real-world issues it could lead to. The introduction would be stronger if it included concrete examples of how unfair or biased code completions might cause practical issues in domains like finance, education, or hiring systems.

For dataset construction, the template-based design might be too synthetic and may not fully represent how code generation happens in real developer environments. This could limit the benchmark’s practical impact.

In Section 3.2, even though CCoT introduces contrastive reasoning, it still relies on the same LLMs for bias evaluation, which may introduce circularity or hidden assumptions regarding the model’s reasoning reliability. In addition, the explanation of how assertions are automatically generated is insufficiently detailed. Please clarify how false positives are avoided and how logical equivalence is ensured in non-trivial code functions.


The Code Bias Score is easy to understand, but it only measures bias presence, not bias severity. A more nuanced metric might capture differences in how serious or impactful a bias is.

Table 1 show thatimplicit bias amplification when removing sensitive attributes), but the paper does not sufficiently explore why this happens or what implications it has for LLM training and deployment.

The findings (that LLMs show social bias) are somewhat expected. The paper doesn’t clearly explain how these results should influence future model development or fairness interventions.

The notion of “social bias in code” needs clearer operationalization. Is the bias measured in logical structures, variable naming, control flow, or comment semantics? A more granular taxonomy would strengthen the conceptual clarity.

**Questions:**

Q1. What is the real-world motivation and impact of studying fairness in LLM-generated code especially in code completion task?

Q2. How does the dataset ensure realism and relevance to actual developer behavior and code completion contexts?

Q3. Since CCoT uses LLMs to generate assertions and analyze bias, what mechanisms are in place to ensure objectivity, avoid self-validation, and maintain reliability in detection?

Q4: Does CBS capture the degree or severity of bias, and if not, how can it be extended to provide a more nuanced and quantitative understanding of bias impact?

Q5. Why does implicit bias amplification occur after removing sensitive attributes, and what are its implications for LLM fairness training and deployment?

**Details Of Ethics Concerns:**

N.A.

---

### Official Review · Reviewer_9Kxw · 2025-11-07

**Soundness:** 1
**Presentation:** 1
**Contribution:** 1
**Rating:** 2
**Confidence:** 4

**Summary:**

This works presents a novel benchmark for evaluating social biases in code completion. First, the authors construct a dataset of code prompts designed to elicit potentially biased completions. Then they evaluate different large language models (LLM), by finding the Code Bias Score (CBS) on the aforementioned dataset. Finally, the authors propose a method for detecting biased code through including assertions in code snippets.

**Strengths:**

The work recognizes the important problem of bias in code generation. It also makes a relevant distinction between code completion and full code generation, which better reflects real-world LLM-assisted programming scenarios.

**Weaknesses:**

1. The dataset used in this work is provided without any detailed explanation of its design. Without a thorough and principal description of the creation of the dataset, it becomes difficult to assess the generalizability of the results presented in this work. Thus, it is hard to argue that any empirical results can be generalized to real-world scenarios, which also undermines the claim that the proposed CCoT method is more effective than the alternatives.

2. The authors present an evaluation on the CBS score of multiple models. However, these models are outdated and far from the capabilities of state-of-the-art models. This puts into question whether biased code is currently a problem or not. For example, recent reasoning models are the main tools used for code generation

3. Some implications presented in Subsubsection 4.3.1 seem to be misleading. Most notably, the sentence "In the Education label, [...] the few-shot setting (92.68%) also outperforms all comparative methods" seems incorrect, as the Few-shot method shows higher consistency (97.56%). More broadly, the CCoT is the best method only in the "Region", "Race" and "Full Dictionary" settings. In the first and second of these three methods, it is tied with Few-Shot-CoT. This is in contradiction with the statement of the authors that "our method (CCoT) exhibits significant advantages across multiple key label classification tasks".

4. The authors claim that the main disadvantage of the CoT method is that it is computationally costly. However, it is not clear that CCoT solves that problem since the corresponding prompts also ask the model to use a given chain of thought. This can be combined with the previous point to further weaken the position of the proposed CCoT method as an improvement over CoT.

5. The work does not adequately position its approach with respect to prior work in prompting with contrastive chain of thought in LLMs (Chia et al. 2023).

6. The scope of the benchmark in terms of supported programming languages seems to be rather limited.

References:
Chia, Yew Ken, et al. "Contrastive Chain-of-Thought Prompting." CoRR (2023).

**Questions:**

1. Could the authors provide a more thorough description of how you build the dataset?

2. Could the authors report CBS scores of state-of-the-art reasoning LLMs? In a similar vein, would they be able to provide detection accuracy on reasoning LLMs for each of the methods shown in Table 2?

3. Could the authors provide additional insight into why the CCoT fails?

4. Can the authors quantify and compare the cost of each method?

5. Have the authors considered extending the benchmark to multiple programming languages, and if so, does language choice influence observed bias?

---

### Meta-Review · Area_Chair_LwcV · 2026-01-08

**Summary:**

The paper explores an important problem of evaluating social fairness in LLM-generated code, and proposes a new benchmark (CodeBiasBench) along with a detection method (Contrastive Chain-of-Thought, CCoT). While reviewers acknowledge the relevance of studying bias in code completion and appreciate the effort to move beyond purely natural-language-based evaluations, they raised substantial concerns that outweigh these strengths. In particular, reviewers found that the benchmark design lacks sufficient realism and methodological grounding, the evaluation protocol is problematic, and the proposed metrics and detection approach do not reliably measure what the paper claims. As a result, reviewers were not convinced that the empirical findings provide valid or actionable insights into fairness in code generation.

**Reviewer Concerns:**

No rebuttal was posted by the authors, and the concerns stand. A clear and operational definition of fairness in code generation is lacking, making the proposed Code Bias Score difficult to interpret. The reliance on LLMs both for code generation and for bias detection via CCoT introduces circularity and self-confirmation bias. In addition, reviewers noted issues with outdated model choices, insufficient comparison to recent related work, unclear claims about computational efficiency, and multiple presentation and organizational problems. Collectively, these issues remain unaddressed.

**Reviewer Scores:**

No score changes are expected, as no rebuttal or discussion was posted.

---

### Decision · Program_Chairs · 2026-01-26

Reject